

# Can Limits to Growth in the Renewable Energy Sector be Inferred by Curve Fitting to Historical Data?

Kristoffer Rypdal[1]

[1]Department of Mathematics and Statistics, UiT The Arctic University of Norway, 9037 Tromsø, Norway

*Correspondence to:* Kristoffer Rypdal (kristoffer.rypdal@uit.no)

**Abstract.** The paper examines the assertion that limits to growth in the renewable energy sector can be inferred statistically from global historical data for installed capacity of solar and wind power. This claim has been made in the peer reviewed scientific literature and has been subject to considerable media coverage. If true, the harsh implication is that the substitution of fossil fuels by renewables is impossible due to Earth System constraints. It is demonstrated here that rational selection between an exponential and a logistic growth model cannot be made from existing data for the historical evolution of global installed capacity. For global power consumption, the growth limit for a fitted logistic model is so high that the growth is indistinguishable from exponential growth for any practical purpose. It is observed that some regional data show polynomial, rather than exponential, growth. But there are no signs of levelling off to a finite limit, and it is suggested that the difference between global and regional growth patterns may be that increasing technology transfer to undeveloped areas gives an extra impetus to the global growth. If this is the case we may see slower than exponential growth in the future without a definite growth limit. These overall negative results regarding drawing definite conclusion from statistical considerations lead to the conclusion that, even though statistical methods are indispensable in energy planning, they are not a substitute for physical and economic modelling.

*Copyright statement.* TEXT

## 1 Introduction

It is generally recognised that economic growth in most sectors finally will have to come to an end due to the constraints imposed by planetary boundaries and that we need a new paradigm in Earth System science that integrates the physical, biological, economic, social and cultural forces (Donges et al., 2017). Nevertheless, energy production and distribution is the sector on which everything else depends, and despite steady advances in energy efficiency, the growth of the world economy relies on continuing growth of energy consumption. Without a massive deployment of carbon capture and storage (CCS), the target of global warming below $2°$ C from preindustrial temperatures requires radical reduction of coal in electricity production over next decades (IPCC, 2014). At present there is doubt about the technical and economic feasibility of capturing and storing $4 \, Gt \, CO_2$ annually by 2040. This is more than 10% of the emissions from fossil fuels and industry, and will require thousands of



large-scale CCS plants (Le Quéré et al., 2016; Global CCS Institute, 2015). The known reserves of conventional oil and natural gas will set strict limitations to the growth in the consumptions of these fuels, although fracking technologies will extend their time window somewhat (IPCC, 2014). In theory, large scale implementation fourth generation nuclear power with a total reformation of the nuclear industry and the national and international regulatory systems, could buy some time (Makabe, 2017),

but the political feasibility of such a project is highly questionable. Most integrated assessment models (IAMs) used in IPCC (2014) include optimistic assumptions on implementation of CCS, but still the majority of such models conclude that a future growth rate of 1-2% in world gross domestic product constrained by the two-degree global temperature target will require that solar and wind power production and consumption continues to grow at present rates without fundamental constraints (Nordhaus, 2013). A fourth option, geoengineering, has not been seriously implemented in the IAMs yet, since these are the

least matured set of technologies, and associated with profound ethical issues. Thus, the rather depressing state of affairs is that the prospect of meeting the IPCC temperature targets rests on the economic feasibility of accelerating growth of world-wide, large-scale deployment of at least one of four classes of technologies; CCS, 4th generation nuclear, geoengineering, or renewables, and it is by no means obvious that any of them can meet the world's demand for clean, safe, and affordable energy.

The outlook is not brightened by the existence of deep scientific controversies, often with ideological and political overtones,

regarding the prospect of each of those of technologies. The peer-reviewed scientific literature, as well as comprehensive reports from sources like the International Energy Agency (World Energy Outlook, 2016) and Greenpeace (Energy revolution, 2015) diverge substantially in their outlooks for the world's energy future. This type of scientific controversy is rooted in intellectual bias and/or lack of knowledge, and by logical necessity; a considerable fraction of the published results must be false. This is a serious problem for energy science and for our society. If energy science shall fulfil the ambition of providing useful guidance

for energy planners, investors and policymakers, falsification of the incorrect results must be given high priority. This is the backdrop of the present paper. It has focus on the possible constraints on the growth of the intermittent power sources; solar and wind. Consumption of hydropower and traditional bioenergy are considerably larger at present, but their growth potential is almost exhausted. For hydro this is true in the developed world, while some developing countries still have large unexploited resources. The extent of these, and therefore the growth potential, are well known. Solar and wind represent proven technologies

of a certain maturity, but their intermittency represents an obstacle that is held by some to be a fundamental constraint to further growth. These and other constraints have been discussed in many recent papers, e.g., Moriarty and Honnery (2011); Dale et al. (2011); Hall et al. (2014); Davidsson et al. (2014). These outline a large number of restraining factors that that may slow, and possibly halt growth of renewable energies, whose low energy return on investment may negatively impact general economic growth. However, the majority of these papers do not present balanced treatments of impeding and accelerating factors, and do

not make quantitative, integrated assessments of all these in a setting where energy markets develop in a world with effective implementation of climate change policies, including global pricing of carbon emissions.

This landscape of huge uncertainty in projections for the market of renewables, and the complexity of modeling them, have led some authors to search for signs of stagnating growth in historical data for deployment of the fastest growing renewable energy technologies. Notably, Hansen et al. (2017) attempt to make a model selection between exponential and logistic growth

of wind and solar power based on standard curve fitting to historical data. The logistic growth curve has the form of a sigmoid,





where the initial exponential growth converges to a maximum value due to a nonlinear saturation mechanism. They conclude that the logistic curve generally yields "better fit," and that there is a statistically significant decline in the relative growth rate, signifying slower-than-exponential growth. The fitted logistic curves indicate a stagnating optimum level of installed wind- and solar capacity not much higher than twice today's capacity, which effectively would remove solar and wind power from the list of potentially "life-saving" technologies. The harsh implications of these projections make it worthwhile to examine their substance in some detail, and to explore whether conclusions of this nature can be drawn from historical data via application of more rigorous methodologies.

This topical paper is structured as follows. In Section 2 the inadequacy of usual least mean square fitting for models with multiplicative noise is explained and illustrated by an analysis of the growth curve for global installed solar power. The stochastic equations for exponential and logistic growth with multiplicative noise are then formulated, and an alternative least mean square fitting method, where the logarithms of these models are fitted to the logarithm of the data, is shown to be one that responds to the entire time series, not only the greater values at its end. The section also discusses the issue of model selection, and presents an intuitive method for assessing the model uncertainty; the uncertainty in the estimates of the model parameters from the data. This method is employed in Section 3 to data for global installed capacity of the sum of solar and wind power, and it is shown that the logistic model makes little practical sense for the growth of global *consumption* of combined solar and wind power, because the modelled logistic growth saturates at unrealistically high levels. In this section, the difference between global and regional growth is illustrated by an example for wind power in Europe, which grows slower than exponential, but without signs of saturation. Section 4 presents a summary of the results and a discussion of the findings in a philosophy of science perspective, emphasising the basic principles of hypothesis testing and the need for scientific journals to publish corrections of false positives, known as type 1 statistical errors.

## 2 Methods

Standard curve fitting is an example of regression where one estimates the parameters (regression coefficients) $\boldsymbol{\alpha}$ of a statistical model of the form;

$$y = f(t; \boldsymbol{\alpha}) + \epsilon, \tag{1}$$

where $t$ is the *predictor variable* (in our case; time), $\epsilon$ represents the "random" or "unexplained" part of the *response variable $y$* (e.g., installed capacity), and $f(t; \boldsymbol{\alpha})$ is some specified function. Suppose we have $n$ observations $\{(t_i, y_i)\}$, $i = 1, \ldots, n$ of the predictors and the response variable, then *regression* means to find the regression coefficients $\boldsymbol{\alpha}$ such that the set of residuals

$$\{r_i \equiv y_i - f(t_i; \boldsymbol{\alpha})\}, i = 1, \ldots, n$$

is minimised in a metric (norm) to be specified. A commonly used metric is the *least square* objective function

$$Q_2(\boldsymbol{\alpha}) \equiv \sum_{i=1}^{n} r_i^2 = \sum_{i=1}^{n} |y_i - f(t_i; \boldsymbol{\alpha})|^2. \tag{2}$$





## 2.1 Multiplicative noise and fitting to log-data

Minimising the least-square deviation to yield the best estimate $\boldsymbol{\alpha} = \hat{\boldsymbol{\alpha}}$ often leads to the best visual fit of the curve (graph) of $f(t; \hat{\boldsymbol{\alpha}})$ to the data, but for data where the fluctuation level $\Delta y_i = |y_i - y_{i-1}|$ is proportional to $y_i$ (multiplicative noise), this metric will not provide the best *model* for the growth, since the estimated model parameters will be very sensitive to the

random fluctuations of the larger data points in the time series. This sensitivity is clearly demonstrated in Fig. 1(a) where the full red and blue curves are least square fits of an exponential and a logistic growth model, respectively, to a time series of global installed capacity of solar power from 1997 until 2015 (black dots). The green dot represents the new capacity installed in 2016, and the dashed lines represent the new fits when this additional data point is included. The interesting observation is the remarkable effect inclusion of one extra data point has on the logistic fit. A more relevant quantity to minimise is the

mean square of $z - \ln f$, where $z = \ln y$, since the fluctuations $dz = dy/y$ will have magnitudes that no longer are proportional to $y$ (additive noise). The model to fit is then $\ln f(t, \boldsymbol{\alpha})$; for the exponential model this reduces to fitting a straight line to the log-data, and for the logistic function fit, it corresponds to fitting a function which has the slope of the initial relative growth rate for $t \ll t_s$ and a zero slope for $t \gg t_s$, where $t_s$ is the inflection point of the logistic growth curve. The logistic model and exact meaning of $t_s$ is explained further in Sect. 2.3. In Fig. 1(b), the fits are made on the logarithm of the data, and we observe

that the extra data point has almost no effect (the dashed curves are almost on the top of the full ones). It is tempting to interpret Fig. 1(a) in support of the exponential model, since the additional data point does not change this model much, but this fitting method does not let the fitted models allow the "natural" multiplicative variability which characterises an expanding economy. The fitting method applied in Fig. 1(b), on the other hand, lets the models accept this variability, and therefore the additional data point does not require a significant change of the parameters of either model.

## 2.2 Exponential growth and the Black-Scholes stochastic equation

The rationale for operating on the logarithm $\ln y$ rather than on $y$ can be seen from the canonical Black-Scholes (BS) stochastic differential equation (SDE) for asset prices, which is a general description of any continuous-time variable stochastic process $y(t)$ that grows at a rate $\mu y(t)$ and is subject to random increments $y(t) \sigma \, dB(t)$. Since the growth rate is proportional to the asset price $y(t)$ this term contributes to exponential growth of $y(t)$, while the stochastic term gives rise to price fluctuations.

Since the magnitude of the fluctuations are proportional to $y(t)$ this is an example of *multiplicative noise*. The multiplicative noise in $y$ reduces to *additive noise* in $z = \ln y$. The equation takes the form,

$$dy = \mu y \, dt + y \sigma \, dB(t), \tag{3}$$

where $B(t)$ is the Wiener process (also called Brownian motion), $\mu$ represents the general economic growth rate and $\sigma$ measures the strength of the price fluctuations. The essential properties of the Wiener process is that the increments $dB(t)$ are identical

and independently distributed (i.i.d.), from which it also follows that the distribution is Gaussian. With discrete time steps, e.g., a time series of annual data, the Brownian motion reduces to a Gaussian random walk process. The equation for the logarithm




**Figure 1.** Black dots show installed solar power capacity globally for the years 1997-2015 (year one in the figure is 1997), while the green dot is capacity installed by the end of 2016. In (a), the red full curve is a fit to the black dots of the exponential model $f(t; y_0, \mu) = y_0 \exp(\mu t)$, while the red dashed is the corresponding fit to the black dots plus the green dot. The blue full curve is a fit to the black dots of the logistic model $f(t; y_m, \mu, t_s) = y_m/(1 + \exp[-\mu(t - t_s)])$, while the blue dashed is the corresponding fit to the black dots plus the green dot. In (b), the models $\ln f(t; y_0, \mu) = \ln y_0 + \mu t$ and $\ln f(t; y_m, \mu, t_s) = \ln y_m - \ln(1 + \exp[-\mu(t - t_s)])$ have been fitted to the logarithm of the data points, and the result presented in a logarithmic plot. As in (a), the full curves are fits to the black dots, while the dashed curves are with the green dot included (here the dashed curves are barely visible because they are almost on top of the full curves).

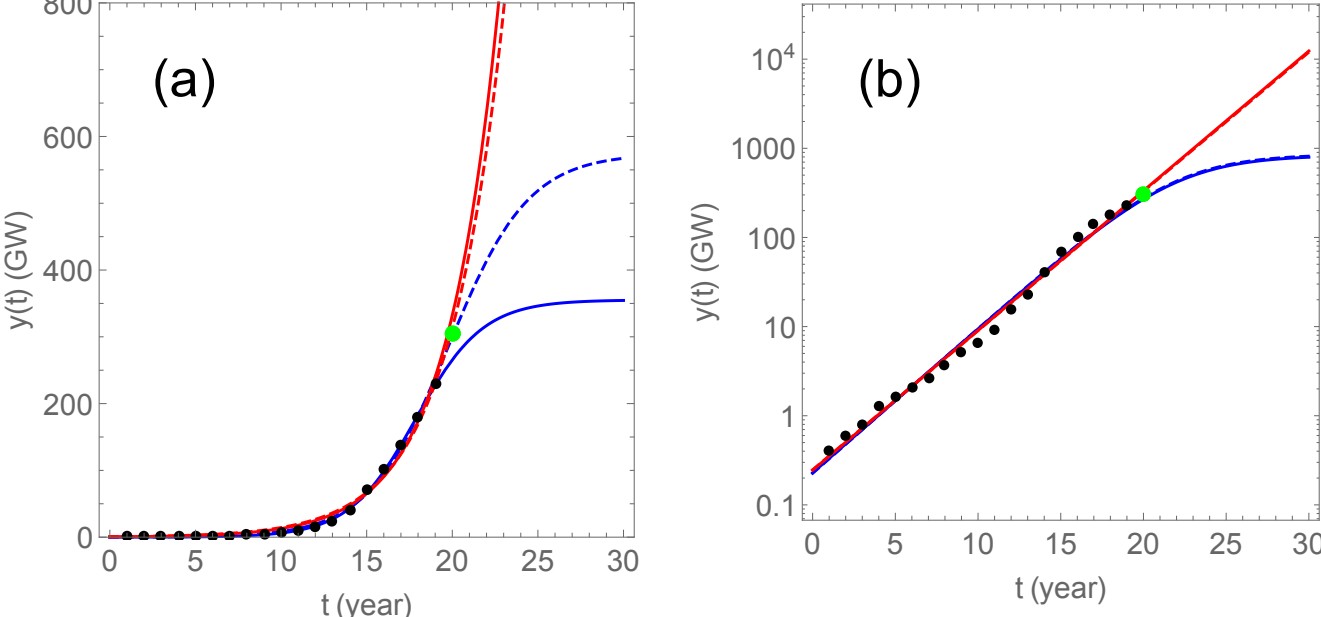

$z = \ln y$ is,

$$dz = (\mu - \sigma^2/2)\, dt + \sigma\, dB(t),\qquad (4)$$

The non-intuitive term $-\sigma^2/2$ in the drift coefficient in Eq. (4) arises because the equation is an SDE for the stochastic process $y(t)$. For a change of variable like $z = f(z) = \ln y$, we have Itô's first lemma, which states that if $y(t)$ satisfies Eq. (3), and $f(y)$ is a twice differentiable function, then the stochastic process $z = f(y)$ satisfies the SDE $dz = [\mu f'(y) + (1/2)f''(y)\sigma^2 y^2]\, dt + \sigma\, dB(t)$. For $f(y) = \ln y$ this equation reduces to Eq. (4). It implies that the stochastic forcing gives rise to an additional drift. The solution to Eq. (l4) is geometric Brownian motion (gBm);

$$y(t) = \exp[z(t)] = y_0 \exp\left[(\mu - \frac{\sigma^2}{2})t + \sigma B(t)\right],\qquad (5)$$

The deterministic factor $\exp(\mu - \frac{\sigma^2}{2})t$ grows exponentially and the probability density function (PDF) of this stochastic process is skewed and log-normal. The expected value of this distribution grows linearly as $\mathbb{E}[y] = y_0 \exp[\mu t]$ and the variance as



**Figure 2.** (a) Realisation of a Gaussian white noise process. (b) Realisation of the geometric Brownian motion with zero drift ($\mu = 0$), i.e., $x(t) = \exp B(t)$, where $B(t)$ is a realisation of a Wiener process (Brownian motion). Note that $x(t)$ is strictly positive.

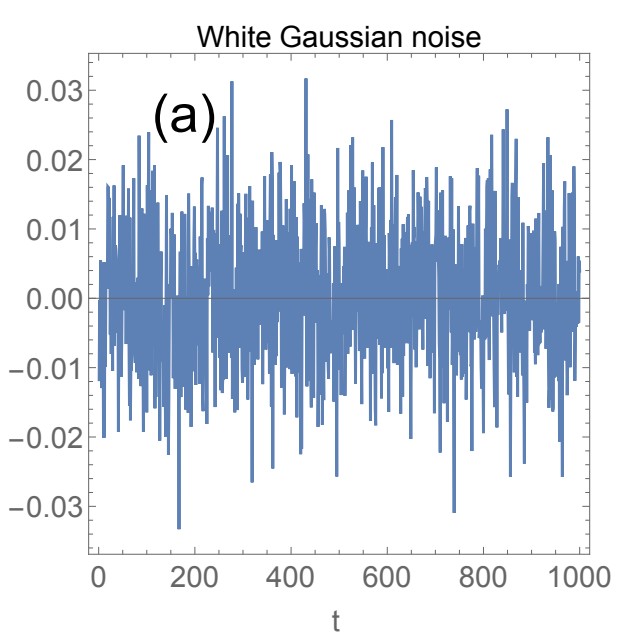

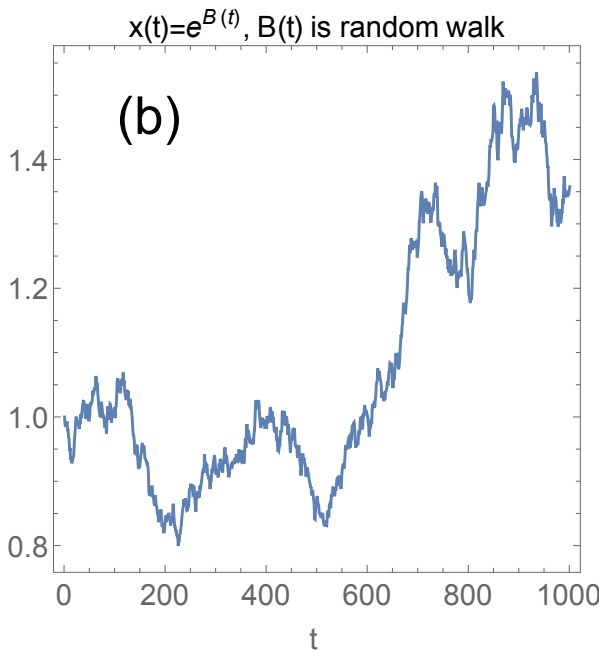

$\mathrm{Var}[y] = y_0^2 \exp[2\mu t](\exp[\sigma^2 t] - 1)$. This variance represents the statistical uncertainty associated with market fluctuations in an exponentially expanding economy.

## 2.3 A stochastic equation for logistic growth

The debate over the growth of installed capacity or consumption of renewable energy is concerned with whether the deter-
5  ministic factor $y_0 \exp[(\mu - \frac{\sigma^2}{2})t]$ should be replaced by a function exhibiting limited growth, such as a logistic function. In making this assessment, however, one has to take into consideration the nature of the sources of statistical uncertainty, which for exponential growth is represented by the multiplicative noise factor $\exp[\sigma B(t)]$. While standard curve fitting is based on the assumption that the statistical error is additive random (white) noise, the actual market fluctuations is more accurately represented as multiplicative, autocorrelated process which is the exponential of the Wiener process. Realisations of the two types
10  of processes is shown in Fig. 2. Standard curve fitting assumes that the noise in Fig. 2a is added to the deterministic growth. Black-Scholes theory assumes that the deterministic growth signal is multiplied by the noise in Fig. 2b.

Unfortunately, the factor $\exp[\sigma B(t)]$ is not an accurate description of the noise for times greater than $t_s$ for logistic growth. We can see this from generalising the Black-Scholes equation to a stochastic logistic growth model (SLGM) (Capocelli and





Ricciardi, 1974);

$$dy = \mu y (1 - y/y_m)\, dt + y\sigma\, dB(t). \tag{6}$$

Without the stochastic forcing term, the solution to this equation is the logistic function

$$y_L(t; y_m, \mu, t_s) = \frac{y_m}{1 + \exp\left[-\mu(t - t_s)\right]}, \tag{7}$$

which has the shape of a sigmoid. Here, $\mu$ is the initial exponential growth rate, $y_m = \lim_{t \to \infty} y(t)$ is the asymptotic limit to the growth, and $t_s$ is is the time where $dy_L/dt$ has its maximum value, which is also the time at which $y(t_s) = y_m/2$. Hence, $t_s$ is a characteristic time for saturation of the logistic growth. Here $t_s$ is related to the initial value $y(0)$ through the relation $y(0) = y_m/[1 + \exp(\mu t_s)]$.

For $t \ll t_s$, Eq. (6) is well approximated by Eq. (3), and the gBm of Eq. (5) offers an accurate description. However, after saturation of the growth (for $t \gg t_s$), one should rather linearize Eq. (6) around the stable fixed point $y = y_m$. By introducing the new variable $\tilde{y} = y - y_m$, assumed to remain small compared to $y_m$, the linearised stochastic equation reduces to the Langevin equation,

$$d\tilde{y} = -\mu \tilde{y}\, dt + \sigma y_m\, dB(t), \tag{8}$$

whose solution is known as a stationary process with standard deviation $y_m \sigma / \sqrt{\mu}$. The full stochastic equation for $z = \ln y$ becomes

$$dz = \left[ \mu \left( 1 - \frac{\exp[z]}{y_m} \right) - \frac{\sigma^2}{2} \right] dt + \sigma\, dB(t). \tag{9}$$

In physics, the Langevin equation was first used to describe the motion of a Brownian particle suspended in a fluid. There $\tilde{y}$ is a the velocity of the particle, $\mu$ is the friction coefficient, and $\sigma y_m\, dB(t)$ represents the stochastic acceleration due to the random kicks from collisions with the thermal molecules of the fluid. The solution to this equation is the Ornstein-Uhlenbeck (OU) stochastic process. The counterpart for a discrete-time process is the first-order autoregressive (AR(1)) process. The OU process is stationary with zero mean and variance $\mathbb{E}[\tilde{y}^2] = y_m^2 \sigma^2 / \mu$. The autocorrelation function is $\mathbb{E}[\tilde{y}(t+\tau)\tilde{y}(t)] = \exp(-\mu\tau)$.

### 2.4 Are there better methods for estimation of model parameters?

During the phase of near-exponential growth (the period we have observations), $z(t)$ approximately satisfies Eq. (4), and the solution is $z(t) \approx (\mu - \sigma^2/2)t + \sigma B(t)$. Hence, for discretised time, the noise is a random-walk process $B(i)$, and not the Gaussian white noise increment process $\varepsilon_i = B(i+1) - B(i)$. The strongest results from standard regression theory are based on the assumption that the noise is a series of identical and independently distributed random variables. Hence, at least from a theoretical viewpoint, there is a case for casting the models into a form where the noise has this character. The BS and the SLGM for the logarithm, Eqs. (4) and (9), both have the general form

$$dz = F(z; \boldsymbol{\alpha})\, dt + \sigma\, dB(t). \tag{10}$$





The discrete-time version of this equation of this equation, with $t = i = 1, 2, \ldots, n$ has the form

$$z_i - z_{i-1} - F(z_{i-1}; \boldsymbol{\alpha}) = \sigma \varepsilon_i, \tag{11}$$

where $\varepsilon_i$ is the random variable drawn from a normal distribution with zero mean and unit variance; $\varepsilon_i \overset{d}{=} \mathcal{N}(0,1)$. A least square fit of this model would be to find the parameters $\boldsymbol{\alpha}$ that minimise the mean square

$$\sum_{i=1}^{n} [z_i - z_{i-1} - F(z_{i-1}, \boldsymbol{\alpha})]^2,$$

and hence the magnitude $\sigma$ of the noise term in the stochastic growth model. Thus, this method effectively fits the derivative of the deterministic solution to the derivative (the differences) of the data. A weakness is that an additive constant remains undetermined, and the superiority of this method over the method of making a least square fit of $z(t) = \log y(t)$ to the log-data is based on the random-walk assumption of the fluctuations. This assumption does not hold very well on the inter-annual time

scales for the type of data considered in this paper. From Fig. 1(b) we observe that the residuals obtained from subtracting $z(t)$ from the log-data time series vary relatively smoothly from one year to the next, but sampled on five years intervals they may be consistent with a random walk. Thus, performing the least-square estimation on $\log y$ rather than on $y$ itself, seems to be important for the estimation of model parameters which takes into account the multiplicative nature of the noise. On the other hand, the improvements by making fits on the derivative are not that obvious, and we shall not employ that method in the

remainder of the paper.

The smooth appearance of the log-residuals on annual scales has important implications for the statistical significance of the downward trend of the relative growth rate claimed by Hansen et al. (2017). The relative growth rate is defined as the slope of the log-data curve, $y'/y = z'$, and is constant in time for exponential growth. Hansen et al. (2017) make a linear regression to the differences $\Delta z_i = z_i - z_{i-1}$, $n = 1, \ldots, 19$ and estimate negative slope of this trend line which is claimed to be significantly

different from zero. Such significance estimates, however, are only valid if the noise in $\Delta z_i$ on annual scale is a Gaussian i.i.d. process. The statistical significance of the negative slope depends critically on the number of independent data points, and if the fluctuations in $\Delta z_i$ are independent only on time scales longer than five years, there are effectively not more than four such points in the data record, and this is clearly not enough to detect a significant trend in these data.

## 2.5   Model selection

The problem we deal with in this paper is to search for evidence for saturation of exponential growth in time series data that to the first order are well described by an exponential function. More precisely, we try to find criteria by which we can reject the BS model for the growth in favour of the SLGM. The first step is to perform a least-square fit of the logarithms of the exponential and logistic models to the logarithm of the data.[1] The exponential fit appears as a straight line in the log-log plot, and the the sigmoid logistic curve starts out a straight line with slope $\mu$, gradually bending over to a straight line with

zero slope as the growth saturates. The exponential model contains two model parameters $y_0$ and $\mu$, while the logistic model

---

[1] Using the fitting routine NonlinearModelFit in Mathematica





contains three; $y_m$, $\mu$, and $t_s$, and we note that the exponential model is a special case of he logistic since the latter reduces to the exponential in the limit $\mu(t_s - t) \gg 1$. Hence, the logistic model should provide a better fit than the exponential to *any* data set in terms of the standard deviation of the residual $\sqrt{Q_2}$ given by Eq. (2).

The parameter estimation described above is associated with statistical uncertainty, which will be provided by standard fitting routines as the "standard error" of the estimation. This is presumably what is done to obtain the confidence intervals on the logistic fit in Fig. 3 of Hansen et al. (2017). These error estimates are based on the assumption that the observed data can be modelled by Eq. (1),where $f(t; \boldsymbol{\alpha})$ is either the exponential or the logistic function, $\boldsymbol{\alpha}$ the corresponding model parameters, and $\epsilon(t)$ a Gaussian white noise process. It is generally recognised, however, that variables describing the volume of an expanding market is much more adequately described by models of the type Eq. (3) or (6). This means that the estimation of the model

errors (the uncertainty in the model coefficients) must be based on those stochastic models.

    According to Eq. (3) or (6) the residual obtained by subtracting the fitted $z(t)$ from the log-data could be modelled as a Wiener process. By estimating the parameters of this process from the residual data, one can produce an ensemble of numerical realizations of the processes described by Eq. (3) or (6).[2] The distribution of these realisations tell us that the parameters of the fitted model is estimated from a random realisation of the process, and hence that fits to other realisations are just as good

as fits to the observations. This mean that we can produce an ensemble of fitted models (a distribution of model parameters) which provides a *model uncertainty*. If this model uncertainty is of the same magnitude for the two models, and they overlap in the entire time period we have observations, we will not have any means of selecting between the two growth models based on the observation data. In the next section it will be shown how this works in practice.

## 3    Results

In this section the methods outlined above are applied to a few data sets related to the growth of intermittent renewable energy resources used for generation of electric power.

### 3.1    Global installed capacity of solar + wind power

The method of model selection is applied to the annual time series given in Table 1 of Hansen et al. (2017) for the sum of installed global wind power and solar power. This time series contains 19 data points $y(t)$, $t = 1, 2, \ldots, 19$, where $t = 1$

corresponds to 1997 and $t = 19$ to 2015.

    The model fits of the logarithms of the exponential and the logistic models to the logarithms of the data are shown as full curves in Fig. 3a, and the ensemble of realisations of the stochastic differential equations corresponding to fitted models are shown as thin lines. The standard deviation for the residuals are $Q_2 \approx 0.065$ for the exponential and $Q_2 \approx 0.045$ for the logistic, and considering that the logistic has one more model parameter, this difference does not provide evidence that the

logistic model is "better" than the exponential.

---

[2]Using the function EstimatedProcess in Mathematica.

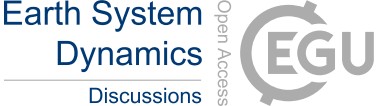

**Figure 3.** In both panels, black dots show integrated installed capacity of the sum of global sol and wind power from 1997 till 2015. (a) The red full curve is the fit of a linear model to the logarithm of the installed capacity, and the black full curve is the fit of logarithm of the logistic model. The red, thin curves are 100 realisations of the BS stochastic process, i.e., solutions to Eq. (3), and the green, thin curves are 100 realisations of the SLGM process given as solutions to Eq. (6). (b) The red and green, thin curves are exponential and logistic fits to the realisations shown in panel (a), respectively.

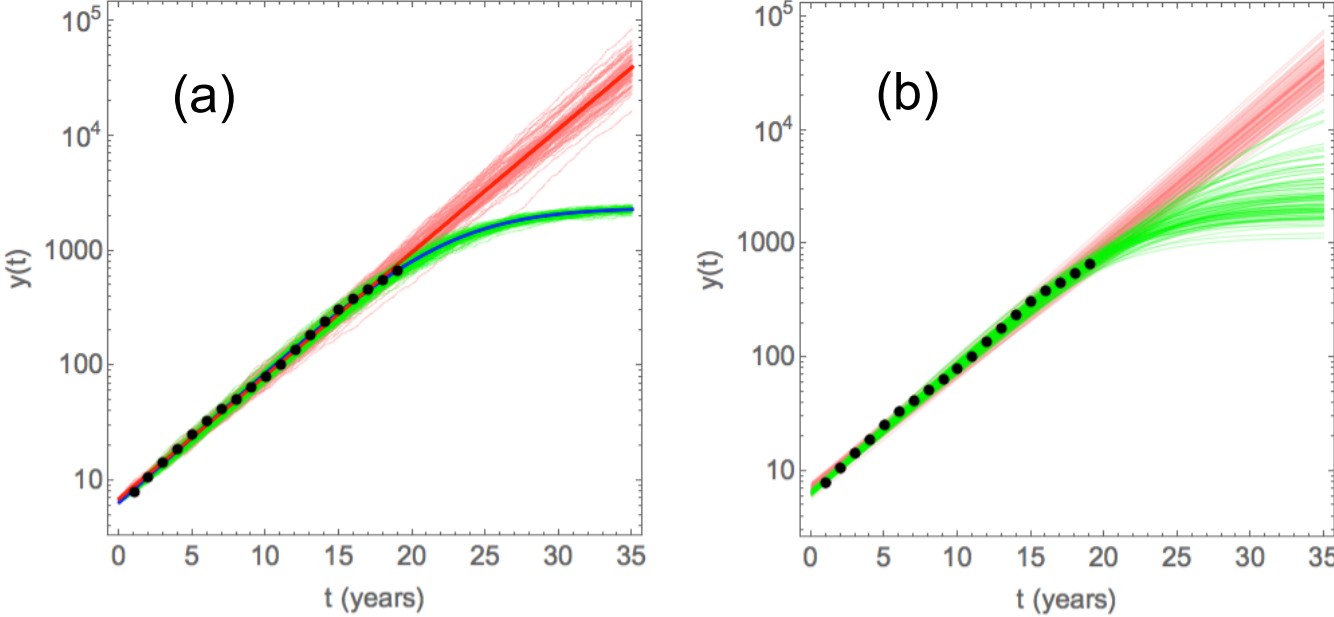

The thin curves in Fig. 3a correspond to 100 such realisations of the BS process described by Eq. (3), and 100 realisations described by the SLGM. The result shows that the range of the realisations overlap in the time interval where we have observation, i.e., both models describe equally well the observed data. Approximately five years from the last observation the two *stochastic uncertainty ranges* start to diverge. One interpretation is that by continuing observations for five years or more

5 it *may* be possible to make a selection between the two models. This conclusion may be too optimistic, however, because the stochastic uncertainty ranges of the two models are centred around fits to the observed data set, while (according to the models) this data set is only one realisation of the underlying stochastic process. In Fig. 3b we have constructed fits to the 100 realisations of the stochastic processes showed in Fig. 3a, and observe considerable larger ranges for the model uncertainties. Fig. 3b indicates that we will have to record data for at least another decade before it can be possible to select between the two

10 growth models. Another conclusion is that the asymptotic limit $y_m$ (the limit to growth) for the logistic growth model is highly uncertain, with a fair chance of being more than an order of magnitude higher than the present installed capacity.



These results suggest that at present it is not possible to select between exponential and logistic growth models for the global installed capacity for solar plus wind power based on historical data for installed capacity, and the uncertainty in the estimates of the parameters of the logistic model makes it useless for inferring a practical limit to the growth.

## 3.2 Global consumption of solar + wind power

In previous sections the analysis was limited to global installed solar and wind power capacity, mainly because these were the data considered by Hansen et al. (2017). A better measure of the growth in these renewable sectors is the total power *consumption*, since this also reflects the growth of implementation of technologies that improve the utilisation of the installed capacity. This includes improvements and expansion of electric grids and better system integration of intermittent power sources. The intermittent nature of solar and wind power is seen by some skeptics as a major hurdle for the continuing growth in the share of these sources in the total electric power mix. Fig. 4a shows the global annual consumption of solar and wind power (the black and red dots) from 1989 till 2015 (BP Statistical Review, 2017). The figure also shows fits of the logarithms of the exponential and logistic models to these data. The full curves are fits to the entire data set for the period 1989 – 2016 (black+red), while the dashed curves are fits to the data for 1989 – 2005 (black dots). The models fitted from the shorter data set are not very different from that arising from the long data set, and indicates that the exponential growth inferred from the shorter set has remained robust throughout the last decade. This may seem counter-intuitive observing the apparent reduced slope during the last three years. A standard fitting of the consumption rather than its logarithm would put greater weight on this feature, while the log-fitting emphasises the trend over the three decades.

Another interesting observation is that the exponential and the logistic fits start to diverge as late as two decades into the future and the logistic model saturates at levels two orders of magnitude higher than the present consumption. An analysis similar to what was done on the installed capacity will show that it is impossible to select between the two models based on the data, but more important, the exponential and logistic models yield predictions that are identical for any practical purpose, since nobody believes that solar+wind will grow by more than two orders of magnitude during the next two decades.

If one believes in extrapolations from historical data, these are very significant results in favour of the exponential growth hypothesis. In fact, since the full curves are above the dashed ones, inclusion of the data from the last decade provides a slightly higher growth rate than inferred from the short data set. These results are particularly interesting because they are at odds with the assertion that the main limiting factor for future growth is not the installed power, but the limitation of consumption due to the intermittent nature of these renewables.

## 3.3 Regional growth

So far, only *global* capacity and consumption have been considered. It is conceivable that a major contribution to the growth of global renewables derives from geographical expansion to new regions where solar and wind power are in its infancy, and hence does not derive from growth in regions with matured technology. If this is the case, one should expect a decline of the growth when these technologies reach a certain level of maturity in all regions of the world. An assessment of these different mechanisms of growth is beyond the scope of the present paper, but Fig. 4b suggests that matured technologies may grow





**Figure 4.** (a) Black and red dots are annual global consumption of the sum of solar and wind power for the period 1989-2015. The red and blue solid curves are a linear fit, and a fit of the logarithm of the logistic model, to the logarithm of these data, respectively. The dashed curves are the corresponding fits to the first 17 years of data (the black dots). (b) Black dots are installed capacity of wind power in Europe in the period 1998-2015. The blue line is an ordinary least square fit of the model $a + bt + ct^2$ to these data. The red curve is an exponential fit to the first ten data points.

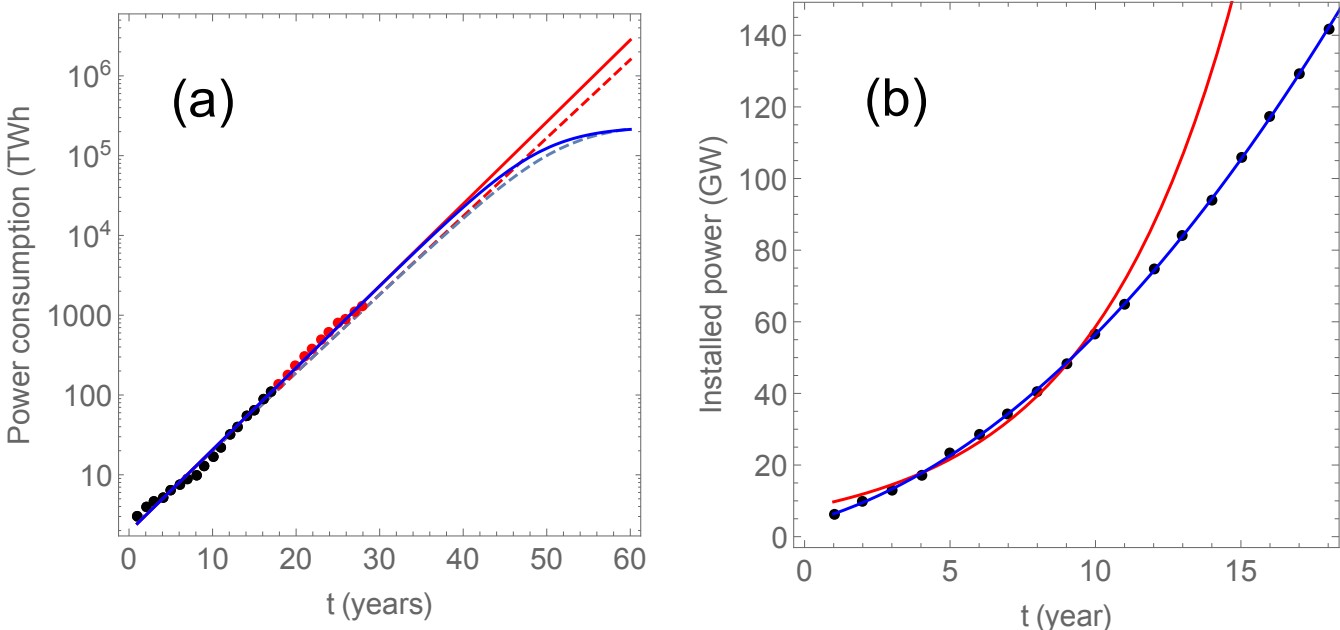

slower than exponential. The black dots in the figure shows the development of installed capacity of wind power in Europe in the period 1997-2015 (BP Statistical Review, 2017). The orange curve is an exponential fit, and the blue curve a fit by a second-order polynomial. The polynomial fit is almost perfect, and indicates slower-than-exponential growth. But it carries no signature of saturation to a stationary limit. This illustrates that if one wants to infer future growth from historical data, one has

5   to look for a broader set of growth models than just the exponential and the logistic.

## 4   Discussion and conclusions

Improper use of statistics in scientific papers is extremely common, and most can be classified as so-called type 1 statistical errors; incorrectly asserting the truth of a hypothesis (false positive). In 2005 Stanford epidemiologist John Ioannidis claimed that "most published research findings are probably false." He found three categories of problems: insufficient assessment of

10  statistical significance; the unlikeliness of the hypothesis; and publication bias favouring novel claims (Ioannidis, 2005). Basic to hypothesis testing is that no statement about nature can be proven true (there is always the possibility that new evidence





will prove it false). Hence, the accepted way of testing a hypothesis (called the *alternative hypothesis*) is to formulate a *null hypothesis* which should be the best possible representation of our prior knowledge of the matter. The test then seeks to falsify the null hypothesis, and if this succeeds, the alternative hypothesis stands stronger, although it is not proven. If the null is not rejected by the test, it has not provided additional evidence in favour of the alternative. Most scientific papers do not even

formulate a null hypothesis, and this is also to some extent the case with Hansen et al. (2017). They actually perform two tests which we will discuss in some more detail:

   (i) The first test is a model comparison between the exponential and the logistic model. In the present paper, two weaknesses of this test have been observed. One is the use of an improper measure of goodness of the fit; the mean square deviation of the data, rather than that of the log-data. This weakness arises from not appreciating the multiplicative nature of the noise in

economic time series (the fluctuations are proportional to the volume). The other weakness is not accounting for the different number of model parameters in the exponential model (two) and the logistic model (three). The latter, which contains the exponential model as a special case, will always give a better fit, and hence come out as the preferred model if the number of fitting parameters is not accounted for. However, information-theoretical criteria exist that penalise model complexity. By applying the most common of these; the Akaike Information Criterion (AIC) or the Bayesian Information Criterion (BIC), the

exponential model comes out as the winner. AIC and BIC are founded on information theory: they offer a relative estimate of the information lost when a given model is used to represent the process that generates the data. In doing so, it deals with the trade-off between the goodness of fit of the model and the complexity of the model. In the most general form they use the maximum of the likelihood function of the model, but assuming for simplicity that the residual is Gaussian and i.i.d., we have $AIC = 2k + n \ln Q_2$, and $BIC = k \ln n + n \ln Q_2$, where $k$ is the number of model parameters and $n$ is the number of

independent data points. The preferred model is the one with the minimum AIC or BIC value. The number of data points in the solar + wind installed capacity is 19, while the number of *independent* points can be as low as 4. In either case, the AIC and BIC values are lower for the exponential model ($k = 2$) than for the logistic model ($k = 3$). The use of such criteria is debatable for the small data sets, but the exercise demonstrates that mean square deviation cannot be used as a model selection criterion.

   (ii) The second test is a hypothesis test that seeks to establish the existence of a negative trend in the relative growth

rate $z' = y'/y$. By using standard criteria for statistical significance, the null hypothesis to be tested is that $z'$ has the form $z_i - z_{i-1} = c + \sigma \varepsilon_i$, where $\varepsilon_i$ are Gaussian and i.i.d. random variables. This null hypothesis is rejected by the test, but it is not a null model that reflects our prior knowledge about the fluctuations in the data. The data exhibit strong dependence on the inter-annual scale, and when the number of independent data points are used ($n \approx 4$), rather than the 19 points in the series, the null hypothesis is not rejected, and the trend does not appear as statistically significant.

Both tests discussed above are examples of very commonly occurring type 1 errors. Correcting them leads to the "negative" conclusion that nothing can be concluded about the limits of growth in the sectors encompassing solar and wind power on the basis of historical data for installed capacity or power consumption. This "boring" conclusion, and the justification for it, are still important to convey. Some readers of the peer-reviewed literature, in particular those with strongly biased views against the future of renewables, will embrace "results" like those presented by Hansen et al. (2017), and accept them as proven scientific

facts. This is of course unfortunate for the advancement of energy science. On the other hand, the majority of readers would



probably *not* trust it, based on a common-sense assessment. After all, who believes that one can predict the future based on tiny deviations from an exponential curve in observation data? However, when such conclusions are presented in the peer-reviewed literature in a superficially convincing statistical language, it inevitably contributes to a general distrust in statistics. This is also unfortunate for the science, and does not contribute to the development of effective strategies for development of the world's
5  energy systems.

*Code and data availability.*  The Mathematica notebook containing the computations is available by request to the author. The data used in Figs. 1, 3, and 4 are found in the reference BP Statistical review 2017.

*Competing interests.*  The author declares that there is no conflict of interest.

*Acknowledgements.*  The author is grateful for useful discussions with Martin Rypdal.



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
