# Peer review of "Can Limits to Growth in the Renewable Energy Sector be Inferred by Curve Fitting to Historical Data?"

_Earth System Dynamics, 2017_

## Referee Comment (RC1) · Anonymous Referee #1 · 13 Nov 2017

The paper by Rypdal presents an important test of recent claims of limits to growth in renewable energy sector that are relevant for mapping out the option space for mitigating greenhouse gas emissions and, hence, future trajectories of the Earth system in the Anthropocene. The presented research shows that based on current data on renewable energy capacity or consumption, logistic growth cannot be distinguished from exponential growth and points out critical methodological flaws in earlier works on the topic.

Since the journal Earth System Dynamics is not focussing on energy science alone, a broader framing and discussion of the problem addressed and the results obtained in the context of Earth system dynamics would be desirable, particularly referring to the topic of the Special Issue "Social dynamics and planetary boundaries in Earth system

modelling". The pedagogical structure and appeal of the paper is useful, even though it is up to debate whether this is an appropriate style for a research article in Earth System Dynamics.

Major issues:

* Sections 3.2 and 3.3: The presentation would be more consistent and convincing if a detailed model uncertainty analysis would be presented for these cases as well as it is done in Section 3.1.

Section 4, p.13, line 27 ff: The statement "The data exhibit strong dependence on the inter-annual scale, and when the number of independent data points are used (n=4), rather than the 19 points in the series, the null hypothesis is not rejected, and the trend does not appear as statistically significant." needs further explanation. How is the number of independent data points computed here?

* Reference should be made to standard literature on model selection where appropriate in the paper to help interested readers finding more in-depth information.

* Please make the data analysis code available as Supplementary Information to ESD.

Minor issues:

* I would find the figures easier to read if time were indicated in years AD (e.g., 1980 to 2015) instead of years from the beginning of the series (e.g., 0 to 35)
* * *

---

## Author Comment (AC1) · 14 Nov 2017

Thanks for a useful and constructive review. All comments by the reviewer are timely and will be addressed in the revision if the editor invites a revision. Here is just some very preliminary responses.

A broader framing and discussion in the context of Earth System Dynamics will be included, although my intention still is to keep this paper short. A more extensive list of references in this direction will be included.

The reviewer questions whether the pedagogical structure and appeal of the paper is an appropriate style for a research article in Earth System Dynamics. To some extent I share this doubt, which is why I submitted it primarily as a short communication, only

subsidiarily as a research article. According to ESD guidelines "Short Communications report particularly concise and innovative/controversial perspectives on the Earth system." I don't claim that my contribution is particularly innovative, but it is concise, I believe it is correct, and the subject is particularly controversial in the public as well as the scientific debate.

The uncertainty analysis for the cases considered in Section 3.2 and 3.3 can of course be included, although it will not reveal much new, and will require a couple of additional figures. I suggest that it be included in Supplementary Material, since this is intended to be a short paper.

Section 4, p. 13, line 27 ff. Here only a rough estimate of the number of independent data points can be made, since the sample contains only 19 points. The number of independent points is based on a rough estimate of the autocorrelation time. Details will be presented in the Supplement.

Relevant references from the litterature on model selection will be included.

The codes for the analysis and production of the figures will be made available as a supplementary Mathematica notebook along with all data files used.

In figures, time will be given in years AD.

---

## Referee Comment (RC2) · Anonymous Referee #2 · 20 Nov 2017

This manuscript critically evaluates an important finding of Hansen et al Limits to growth in the renewable energy sector. Renewable and Sustainable Energy Reviews 70,759-774, 2017. Specifically it considers whether the conclusion that wind and solar installation is on a logistic or exponential trajectory. Via some primer information on statistical significance and model selection, it concludes that Hansen et al were incorrect to argue that a logistic and so saturating trajectory best characterises wind and solar deployment.

The manuscript's methods and results are robust. My comments are relatively minor and focus on some of the framing of the research. The paper could be significantly and easily improved with regards some of the claims and context for the research. There is a question regarding the significance of the manuscript as at its core the manuscript

takes a specific issue with a particular study. A revised version could serve as the basis for a broader tutorial type article to engage 'consumers' of studies about energy capacities and strategies with the Hansen et al 2017 article as an example case study.

Specific comments:

P1 "It is generally recognised that economic growth in most sectors finally will have to come to an end due to the constraints imposed by planetary boundaries and that we need a new paradigm in Earth System science that integrates the physical, biological, economic, social and cultural forces (Donges et al., 2017)."

The author cites a single 2017 published paper. This opening statement must be softened, or further defended.

P1 "Without a massive deployment of carbon capture and storage (CCS), the target of global warming below 2âŮę C from preindustrial temperatures requires radical reduction of coal in electricity production over next decades (IPCC, 2014)."

CCS is one possible Negative Emissions Technologies (NET). There are others and there are proposed mixes in which CCS and the more speculative technologies will not play a large part. See another Hansen publication: Hansen et al 2017 ESD 10.5194/esd-8-577-2017.

P2 "Most integrated assessment models (IAMs) used in IPCC (2014) include optimistic assumptions on implementation of CCS"

All the 2°C scenarios involve NETs of some sort. This may be large scale afforestation and soil management not just CCS.

P2 "This type of scientific controversy is rooted in intellectual bias and/or lack of knowledge, and by logical necessity; a considerable fraction of the published results must be false. This is a serious problem for energy science and for our society. "

This is an editorial judgement, but I do not think such statements are necessary. It is not
necessary to elude to bias or ignorance. It is sufficient to highlight the wide difference in scenarios and pathways to avoid dangerous climate change that are being offered. It is not necessary as to speculate what drives them.

P2 "Consumption of hydropower and traditional bioenergy are considerably larger at present, but their growth potential is almost exhausted. For hydro this is true in the developed world, while some developing countries still have large unexploited resources."

This statement required evidential support.

P4 It is tempting to interpret Fig. 1(a) in support of the exponential model, since the additional data point does not change this model much, but this fitting method does not let the fitted models allow the "natural" multiplicative variability which characterises an expanding economy. The fitting method applied in Fig. 1(b), on the other hand, lets the models accept this variability, and therefore the additional data point does not require a significant change of the parameters of either model.

AND

P9 "It is generally recognised, however, that variables describing the volume of an expanding market is much more adequately described by models of the type Eq. (3) or (6). This means that the estimation of the model 10 errors (the uncertainty in the model coefficients) must be based on those stochastic models."

This is an important assumption. Supporting evidence is required that wind and solar deployments are driven by the important factors of an expanding economy and that such economies are more adequately described as proposed.

P12 "The orange curve is an exponential fit, and the blue curve a fit by a second-order polynomial. " The curve in Fig 4b is red not orange.

P13 "Some readers of the peer-reviewed literature, in particular those with strongly biased views against the future of renewables, will embrace "results" like those presented by Hansen et al. (2017), and accept them as proven scientific facts. This is of course

unfortunate for the advancement of energy science."

It is not necessary to speculate as to the beliefs or intentions of people who may or may not read other published research.

---

## Author Comment (AC2) · 23 Nov 2017

I also thank Referee #2 for a constructive report that will be very useful in the revision of the paper. I shall not reply to the specific comments in this preliminary response; I agree with most of them. More detailed response will be given later when I submit the revision. I just would like to point out that both referees seem to favour the development of a broader tutorial type article to with the Hansen et al 2017 article is treated as an example case study. Personally, I think this is a very appealing challenge. But it is somewhat at odds with the short communication style of the original manuscript. However, if the editor encourages a revision in this direction, I will be more than willing to do so.

---

## Referee Comment (RC3) · Anonymous Referee #3 · 28 Nov 2017

This manuscript revisits an earlier assumption that the projected growth in the renewable energy sector can be inferred from global historical solar and wind power. By way of a step-wise statistical primer into the underlying decisions regarding model selection, this manuscript concludes that it is not presently possible to determine if the logistic or exponential model best describes the future.

Overall, I find the core scientific results to be robust. My difficulty and concern in reviewing this manuscript relates to the more non-technical details and how these are presented. The manuscript continually attacks one study and often uses a writing style that, I think, undermines rather than inspires confidence in your results and conclusions.

My recommendation to the Editor – scientific issues need to be addressed before pub-

lication. An optional but not-required suggestion: the manuscript would greatly benefit from heavy editing by the author – add more scientific relevance and applicability to future studies, and remove inflammatory remarks.

Response is therefore broken into 2 parts:

Scientific major issues:

1. Why are only 4 of the 19 points independent (p.14 line 21 and elsewhere); Is this related to the text on p.8 line 21-24?

2. Possibly related to above, how would the random exclusion of data points inform the error estimation and associated model selection

3. Question relevant to your conclusion – what would be inferred if only the first or second 10 year data series of Fig. 1 was utilized – see p.10 line 9

Scientific minor issues:

4.Consumption being preferrable to installed capacity (Sec. 3.2)? I don't understand the relevance, as 2015 consumption of renewables is a very small fraction ($\sim$0.1 of 18 TW total primary; $\sim$0.1 of 2.0 TW of electricity consumption; Fig. 4A). "These results [based on consumption rather than installed capacity] are particularly interesting because they are at odds with the assertation that the main limiting factor for future growth is not the installed power, but the limitation of consumption due to the intermittent nature of these renewables" (p.11 line 25). Without a suitable reference to support this claim, my assumption is that the proportion is so small that even a highly variable generation source could generally be incorporated into the system and may only prove challenging as this proportion grows to 10s of percent.

5. Please add an isolated Conclusion paragraph for clarity; Page 2 line 19-22 is also relevant; Page 11 line 1-3 would help inform the reader about what the author's point of the article is.

6. Associated data and code (as markup), descriptive comments, and estimates from the calculation (inline in the code) should be provided in the Supplement

7. Fig. 3 color selection – red and green difficult to distinguish by those that are color blind

8. Fig. 4B does not have an orange curve (p.12 line 2)

9. Add additional references: "...profound ethical issues" p.2 line 10; "...but their growth potential is almost exhausted" p.2 line 23; "...the intermittent nature of these renewables" p.11 line 27

Writing major issues:

10. I think that several parts of the text are inflammatory to even the casual reader. Examples include: "This type of scientific controversy is rooted in intellectual bias and/or lack of knowledge," (p.2 line 18); "...led some authors to search for signs of stagnating growing in historical data," (p.2 line 33); "...since nobody believes that solar+wind will grow by more than two orders of magnitude during the next two decades" (p.11 line 22); "...will embrace "results" like those presented in Hansen et al. (2017) and accept them as proven scientific facts" (p.13 line 33) – these should be softened or edited

11. focus on attacking one study: The inability to contest the results of a review article are appreciated here. Rather than continually attacking Hansen et al. (2017) though, why not take the standpoint of how to improve forthcoming projections or estimates. Otherwise the scientific knowledge it contributes is very limited in scope, as the projection by Hansen et al. (2017) could be right for the wrong reasons. I would much rather see how others can be taught to not to make this same mistake in model selection in the future.
* * *

---

## Author Comment (AC3) · 29 Nov 2017

Major scientific points:

1. Why are only 4 of the 19 points independent?

Response: This question was also asked by another referee. In the revision I will present an additional figure explaining this. It arises from the observation that when considering the residual after subtracting a fitted curve it looks like a smooth curve on time scales shorter than five years, i.e. the autocorrelation time is of this order (observe the slow, smooth oscillation of the data points around the fit curves in Fig. 1). With such a small sample (19 points) this is of course a very crude estimate, but it invalidates the underlying assumption that this is a sample of 19 randomly distributed data points.

[Figure]

2. How would random exclusion of data points inform the error estimation and associated model selection?

Response: I am not sure I understand the point here. I demonstrate theoretically and by example in the manuscript (Fig. 1) that the result depends strongly on the last data points in the time series if the fit is made to the installed capacity itself. However, the result is insensitive to removal of the last data points if the fit is made to the log-data, and this is therefore the preferred method. It should be quite apparent from Fig. 1b that the fits of both models to the log-data are so good that random removal of a few data points would not change the fit parameters by much. Rather than exploring uncertainty by random removal of data points, I have produced Monte Carlo ensembles of realizations of the fitted model stochastic processes that reflect the variance of the residual log-data set after subtracting the fitted model from the log-data (Fig. 3a). Then I have explored model uncertainty by fitting the models to these realizations (Fig. 3b). This is a much more meaningful method of estimating model uncertainty (uncertainty of the fitted model parameters) than random removal of data points. After all, we do have those points, and we don't obtain more information by throwing away meaningful data.

3. What would be inferred if only the first or second 10 year data series of data of Fig. 1 was utilized - see p. 10 line 9?

Response: What would be inferred from using only the first part of the data series was indicated in Fig. 4 for the slightly longer consumption data series. Here, dropping the 10 last data points had almost no effect on the estimated model parameters. Again the fitted models both represent so good fit to the log-data over the entire period that the fit is insensitive to whether using the entire data set or only the first 17 points. What this essentially implies is that exponential growth is an extremely good model for the observed data, while the modeled point of saturation is either very uncertian (Fig.1) or too far into the future to be realistic (there must be physical limits to growth that haven't yet had an observable impact). Since a linear fit through the last few points in Fig. 1b

and Fig. 4 would yield a slightly smaller slope than a fit to the entire data set, a logistic fit would yield a lower growth limit, but the uncertainty would be huge since this local slope could be a fluctuation and not a trend.

Scientific minor issues:

4. Consumption preferable to installed capacity.

Response: Maybe I wasn't sufficiently clear on this point. Actually, the main reason I have used consumption as an additional data set is that this data set is almost a decade longer and thereby demonstrates even more clearly the exponential nature of the growth. But electricity consumption is also a more relevant measure than installed capacity, since it incorporates the effects of advances other than those that are expressed through the nameplate generation capacity. Renewables do not have to constitute a large fraction of the world's electricity production for such advances to take place. A generator's output may vary according to changing conditions at the power plant, in the power grid, or in the electricity market. Nevertheless, it turns out that the exponential growth rates for the two data sets are equal to the second digit, which indicates that the two types of data are roughly equivalent. The reason why the logistic fit to consumption time series stays close to the exponential for a longer period of time into the future is that the series extends further into the past, and hence the result becomes less sensitive to the lower slope of the last few data points.

5-9. These technical issues have also been raised by other referees and will be addressed in the revision.

Writing major issues:

10. Inflammatory remarks?

Response: I would be more than happy to remove inflammatory style, but I have a hard time seeing that the phrases mentioned by the reviewer are inflammatory:

"This type of scientific controversy is rooted in intellectual bias and/or lack of knowledge." This could have been inflammatory if I accuse particular persons or a particular side in the controversy for being biased or unknowledgeable. However, what I am stating is basically that if contradicting results appear in the literature, at least some must be wrong.

". . .have led some authors to search for sign of stagnating growth in historical data.." I can't see anything inflammatory in stating this fact. There is of course nothing wrong in searching for such signs.

". . . since nobody believes that solar+wind will grow more than two orders of magnitude during the next two decades." This is not an inflammatory statement, but it could be formulated more carefully. My point is that the two models yield the same growth up to a consumption level more than two orders of magnitude higher than today. Such a consumption level is beyond reasonable physical limits, and implies that the limits to growth cannot be found from these historical data.

". . . will embrace "results" like those presented by Hansen et al. and accept them as proven scientific facts." I may want to rewrite this entire paragraph, but I still can't see why it is inflammatory to state that some readers may take results published in the peer-reviewed scientific literature as proven scientific facts.

11. Focus on attacking one study, why not focus on how to improve forthcoming projections and estimates? Hansen et al. could be right for the wrong reasons.

Response: Although hypothesis testing and falsification is mainstream in the philosphy of science, it is not in very high esteem in scientific journals. The subject of the present paper is a hypothesis raised in the paper by Hansen et al. (2017). The hypothesis is not that there are specific limits to growth in the renewable energy sector (of course there are), but that these limits can be inferred simply from historical data. I do not agree that I am "continually attacking Hansen et al. (2017)," although it would be improper not to point out where they go wrong. My focus is on pointing at the correct way of making curve-fitting on more or less exponentially growing data and on how to make model

selection. The result of doing this correctly is that we cannot select rationally between logistic and exponential growth based on historical data, and that we cannot conclude anything about the limits to growth from such data. Somehow, the reviewer's view is that this "negative" conclusion implies that the "scientific knowledge it contributes is very limited in scope as the projections by Hansen et al. could be right for the wrong reasons." I find this conclusion very problematic. Would the scientific knowledge contributed have been more valuable if the conclusion were that we CAN conclude from historical data, i.e. are affirmative conclusions more valuable than negative ones? Are the projections of Hansen et al. more right, and my paper less relevant, if their projected limits to growth happen to be quantitatively correct by accident?

---

## Editor Comment (EC1) · A. Kleidon (Editor) · 5 Dec 2017

I received the following comment from Jan Petter Hansen (of Hansen et al. 2013) and post it on his behalf.

Axel Kleidon, Editor

**Comment by J. P. Hansen and D. L. Aksnes to the discussion of the manuscript, "Can Limits to Growth in the Renewable Energy Sector be Inferred by Curve Fitting to Historical Data", by K. Rypdal**

In the paper [1] the authors conclude that the combined wind and solar installed global power capacity show early sign of a logistic development. This was based on an observed decrease (fig 3, inset in [1]) in annual growth rates for the period 1997-2015.

[Figure]

In contrast to the claim in [2], adding data for wind and solar installed capacity from 2016 only strengthen this observation. A decreasing growth rate is a prerequisite for a logistic development.

According to the author [2], such results should not be published because of potential misuse by "those with strongly biased views against the future of renewables". We disagree! As stated in the final sentence of [1] we consider that "the present data are an early warning of a growing gap between expressed ambitions and an actual growth." However, when it comes to predicting future trajectories we fully agree with the author of [2] that there are huge uncertainties and that only future data will provide the answer.

**References**

[1] Hansen, J. P., Narbel, P. A, and Aksnes, D. L.: Limits to growth in the renewable energy sector. Renewable and Sustainable Energy Reviews, 70,759-774, http://dx.doi.org/10.1016/j.rser.2016.11.257, 2017.

[2] Rypdal, K.: Can Limits to Growth in the Renewable Energy Sector be Inferred by Curve Fitting to Historical Data?, Earth Syst. Dynam. Discuss., https://doi.org/10.5194/esd-2017-93, in review, 2017.

---

## Author Comment (AC4) · 5 Dec 2017

I appreciate this comment from J. P. Hansen and D. L. Aksnes (H&A). It helps pinpoint the essence of my critique.

Our disagreement on what one can conclude from the data boils down to whether or not an observed decrease in growth rate for the last few years in the record can be treated as an early warning signal for stagnating growth.

When considering the entire record in Fig.1b one will observe a slow oscillation with period of approximately 15 years around the fitted curves. This is what we would find if we make a polynomial interpolation curve through the data points. If I understand H&A correctly, they interpret the reduced slope for the years 2013-2016 as an early warning

of continuing reduced growth in the future. The logical implication of that interpretation is that if the same approach had been applied some years ago, when the oscillation was in a growing phase, the early warning would have been increased growth. An early warning signal that depends critically on the exact time you choose to detect it is of course useless.

This logical flaw is rooted in a lack of awareness of the importance of distinguishing between the modeled signal and the fluctuations. One of the main points in my paper is that when the signal grows more or less exponentially, standard curve fitting will result in a fitted model that is hypersensitive to the fluctuations in the last part of the data record, and hence in great uncertainties in the estimated model parameters. This problem is solved by making the fit to the log-data rather than to the original data. By making fits excluding and including the data point for 2016 in Fig.1 I wanted to illustrate that standard curve fitting makes the result for the logistic fit extremely sensitive to this single data point, i.e. extremely sensitive to a fluctuation that may not reflect the long-term trend.

As discussed above, it is methodologically flawed to draw conclusions from one year's observation, so I wouldn't care much about whether the inclusion of the last point would give increased or decreased limit to growth in the logistic model. This is illustrated in Fig.1b where the growth limit is essentially unchanged by the last data point when the fit is done to the log-data. Nevertheless, in the light of Fig.1a I am surprised about the following H&A comment:

"adding data for wind and solar installed capacity from 2016 only strengthen this observation. A decreasing growth rate is a prerequisite for a logistic development."

It seems that H&A interpret the 2016 data as an additional point on the downward trend of the instantaneous growth rate. However, Fig.1a shows that, although this last data point is still slightly below the exponential curve, the standard curve fitting method lifts the saturation limit of the logistic curve by 50 percent when this point is included.

Hence, by following the logic of H&A, this point could be interpreted as an early warning of a growing saturation limit, i.e. as a recovery of exponential growth. These conflicting interpretations illustrate that it is impossible to avoid subjectively biased judgments when applying the non-robust methodology of H&A.